# “Nobody Listened”. Mothers’ Experiences and Needs Regarding Professional Support Prior to Their Admission to an Infant Mental Health Day Clinic

**DOI:** 10.3390/ijerph182010917

**Published:** 2021-10-17

**Authors:** Tinne Nuyts, Sarah Van Haeken, Neeltje Crombag, Binu Singh, Susan Ayers, Susan Garthus-Niegel, Marijke Anne Katrien Alberta Braeken, Annick Bogaerts

**Affiliations:** 1Department of Development & Regeneration, Faculty of Medicine, Woman & Child, KU Leuven, 3000 Leuven, Belgium; tinne.nuyts@kuleuven.be (T.N.); sarah.vanhaeken@ucll.be (S.V.H.); neeltje.crombag@kuleuven.be (N.C.); 2Research & Expertise, Resilient People, UC Leuven-Limburg, 3590 Diepenbeek, Belgium; 3University Psychiatric Center, KU Leuven, 3000 Leuven, Belgium; binu.singh@uzleuven.be; 4Center for Maternal and Child Health Research, City, University of London, London EC1V 0HB, UK; Susan.Ayers.1@city.ac.uk; 5Institute for Systems Medicine (ISM), Faculty of Human Medicine, MSH Medical School Hamburg, 20457 Hamburg, Germany; Susan.Garthus-Niegel@uniklinikum-dresden.de; 6Institute and Policlinic of Occupational and Social Medicine, Medical Faculty, Technical University of Dresden, 01307 Dresden, Germany; 7Department of Child Health and Development, Norwegian Institute of Public Health, 0473 Oslo, Norway; 8REVAL-Rehabilitation Research Center, Faculty of Rehabilitation Sciences, Hasselt University, 3590 Diepenbeek, Belgium; marijke.braeken@ucll.be; 9CRIC Centre for Research & Innovation in Care, Department of Nursing and Midwifery, University of Antwerp, 2610 Wilrijk, Belgium; 10Faculty of Health, University of Plymouth, Devon PL4 8AA, UK

**Keywords:** infant mental health, perinatal mental health, resilience, infant-parent interaction, healthcare organization, regulatory dysfunction, holistic health, delivery of health care integrated

## Abstract

Challenges during the perinatal period can lead to maternal distress, negatively affecting mother-infant interaction. This study aims to retrospectively explore the experiences and needs regarding professional support of mothers with difficulties in mother-infant interaction prior to their admission to an infant mental health day clinic. In-depth semi-structured interviews were conducted with 13 mothers who had accessed an infant mental health day clinic because of persistent severe infant regulatory problems impairing the wellbeing of the infant and the family. Data were transcribed and analyzed using the Qualitative Analysis Guide of Leuven (QUAGOL). Three themes were identified: ‘experience of pregnancy, birth, and parenthood’; ‘difficult care paths’; and ‘needs and their fulfillment’. The first theme consisted of three subthemes: (1) ‘reality does not meet expectations’, (2) ‘resilience under pressure’, and (3) ‘despair’. Mothers experienced negative feelings that were in contradiction to the expected positive emotions associated with childbirth and motherhood. Resilience-related problems affected the mother-child relationship, and infants’ regulatory capacities. Determined to find solutions, different healthcare providers were consulted. Mothers’ search for help was complex and communication between healthcare providers was limited because of a fragmented care provision. This hindered the continuity of care and appropriate referrals. Another pitfall was the lack of a broader approach, with the emphasis on the medical aspects without attention to the mother-child dyad. An integrated care pathway focusing on the early detection of resilience-related problems and sufficient social support can be crucial in the prevention and early detection of perinatal and infant mental health problems.

## 1. Introduction

Becoming a mother is a unique life event that can be overwhelming [1]. Important factors that facilitate the transition to motherhood are social support [2,3,4,5], correct and realistic information [4,5], and attention to the psychological changes [4]. Unrealistic expectations, feelings of stress and loss of control, lack of sleep, and lack of adequate health care may compromise this transition [2,5]. A variety of psychological, social, and biological determinants challenge mental health during or after pregnancy [6,7].

Perinatal mental health problems occurring during pregnancy or in the first year following birth, affect approximately 20% of women [8,9], thus making them an important public health issue [10]. According to the primary tenets of infant mental health (IMH), an infant’s mental health cannot be considered separately from that of its primary caregiver [11]. As young children rely on their caregivers for the development of their capacity to self-regulate [12], the quality of parenting is crucial [13]. Infant regulatory problems like excessive crying, feeding, or sleeping problems can negatively influence the mental health of the child in the long term [14] and compromise the mother-infant relationship [15], if not treated adequately.

According to the Developmental Origins of Health and Disease (DOHaD) hypothesis, the time from conception up to the age of two (i.e., ‘the first 1000 days) is crucial for the prevention of chronic diseases and disorders including mental health disorders [16]. Exposure to environmental influences during pregnancy and the postpartum period may have significant consequences on the physical and mental health of the mother and her offspring [17]. Growing up in a safe environment with responsive and nurturing caregivers increases the likelihood of strong infant mental health. IMH is defined as, “*The developing capacity of the child from birth to 5 years old to form close and secure adult and peer relationships; to experience, manage, and express a full range of emotions; and to explore the environment and learn—all in the context of family, community, and culture*” [18].

To prevent the intergenerational transmission of mental health problems [19], it is important to promote mental wellbeing in future parents and their infants [20] and to identify mother-child dyads who need professional support and early intervention [14]. Earlier research has mainly focused on the experiences of mothers who accessed a perinatal mental health service for their mental health problems [21,22]. Mothers have stressed the importance of a supportive environment. And a trusting therapeutic relationship is a key factor in creating a safe space where their needs can be met. Perinatal mental health services were considered critical to address the relationship with their infant [19]. Hence, little is known about the experiences and needs of mothers who accessed an IMH facility because of their infant’s regulatory problems. Given the mutual influences of parent-infant interaction, insight into the experiences and needs of these mothers can add value to the development of care pathways and preventive care of perinatal and infant mental health [23]. Therefore, this study aimed to retrospectively explore the experiences and needs of mothers regarding professional support prior to their admission to an IMH day clinic because of persistent severe infant regulatory problems.

## 2. Materials and Methods

### 2.1. Design

An exploratory qualitative research design was adopted, using the Qualitative Analysis Guide of Leuven (QUAGOL). This method is inspired by the grounded theory approach. This method was chosen to reconstruct the story of the participants on a theoretical level and to analyze the concepts found [24,25]. Face-to-face semi-structural individual interviews explored the experiences and needs regarding professional support prior to participants’ admission to an infant mental health day clinic. 

### 2.2. Participants and Recruitment

This study was set in a semi-residential IMH facility in a psychiatric center in Belgium, which focuses on families of infants with persistent regulatory and/or developmental problems. Details about the phases in treatment and program for the parent-infant dyad are described elsewhere [26]. The study received approval from the Ethical Commission Research of University Hospital/Catholic University of Leuven, Belgium.

Purposive sampling was used to ensure participants met the inclusion criteria. These were: (1) being admitted to outpatient services of the IMH day clinic between 2016–2019 with their infant aged 1 to 24 months at the moment of admission; (2) absence of pre-existing bipolar and psychotic disorders; (3) absence of major depressive or anxiety disorder at the time of recruitment; and (4) minimum age of 18 years. Based on these inclusion criteria, 21 out of 42 mothers admitted to the IMH facility in this period, were eligible. Of these, 13 women participated in the study.

### 2.3. Ethics Statement

Each interview started with reviewing and signing the informed consent document. Participants took part voluntarily and had the opportunity to discontinue their participation at any time during the interview and study process. In case of any distress, the IMH service provided backup. Transcripts were coded to guarantee the confidentiality of the participants. The study was conducted according to the guidelines of the Declaration of Helsinki and the study protocol and accompanying documents were approved by the Ethics Committee Research of the University Hospital (UZ/KU Leuven, Leuven, Belgium).

### 2.4. Data Collection

Eligible women received study information from an IMH facility staff member and were asked whether they wished to participate in an in-depth semi-structured individual interview. Those who agreed were contacted by one of the researchers (TN, midwife; or SVH, clinical psychologist). After agreeing to participate and giving written informed consent, interviews were conducted by two researchers (TN & SVH) between January 2020 and March 2020. The interviews lasted between 50 min and 1 h 44 min (median: 60 min) and were conducted at the participants’ house, except for two interviews. One interview was conducted in a conference room at the university, the other one in a local restaurant, both at the participant’s request. Each participant was interviewed once. In most interviews (N = 9) the researcher and the participant were present. In some interviews, the participant’s partner (N = 1) or children (N = 3) were also present. 

After signing the informed consent document, participants answered a questionnaire developed for this study to assess socio-demographic data and participants’ family structure. The main focus of the interview was on the experiences and needs regarding professional support for mothers prior to admission to the IMH facility and how they recuperated after admission. The interview topics are shown in Table 1. All interviews were audio-recorded and transcribed ad verbatim. 

### 2.5. Data Analysis

Systematic analyses of the transcripts were conducted following QUAGOL [24]. This method is characterized by a repetitive process and team approach. It consists of two parts: (1) preparation of the coding process and (2) the coding process using qualitative data analysis software. The first part aims to determine a conceptual understanding of the research data as a whole. After each interview, a descriptive report was written by both the researchers, and each interview was (re)read. During the reading process, key phrases and passages were underlined and assembled in a narrative review. Next, the first authors (TN & SVH) developed independently a conceptual interview scheme for each interview. Within this scheme, concrete experiences were converted into a list of concepts. A constant comparative analysis was used in which concepts were compared with each other, within one interview and between different interviews. All concepts used in the conceptual interview schemes were listed, evaluated, and discussed by members of the research team (TN, SVH, NC & AB). The conceptual interview scheme was tested by rereading each interview, and the quality of the scheme was critically examined. If necessary, concepts were adjusted or compiled. Next, concepts were linked to relevant fragments of the interviews, which ensures that the description and the characteristics of the concept are grounded in the empirical data. The second part comprises the actual coding process using qualitative data analysis software [25]. In the present study, the NVivo software (NVivo (Version 12); QSR International Pty Ltd. (2018), Melbourne, Australia) was used. Ultimately, we obtained a meaningful conceptual framework to systematically describe the key findings. To ensure the quality of the study, the analysis was triangulated by the research team. Furthermore, there was an open dialogue within the team to discuss assumptions and one’s situatedness within the research to practice positional reflexibility. Data saturation was obtained after nine interviews. The process was continued until data saturation was reached; no new information was obtained from subsequent interviews and thus recruitment was stopped. To confirm saturation, four more interviews were conducted after which no new themes emerged [27].

## 3. Results

Thirteen mothers participated in this study, Table 2 shows the socio-demographic and IMH day clinic characteristics of the mothers and their families. All mothers and fathers were Caucasian and born in Belgium. The mean (SD) age at the time of the interview was 33.6 (4.3) and 34.3 (2.8) years for respectively mothers and fathers. Most mothers (N = 10) lived with a partner, three women were divorced or no longer living with the father of their child. The majority of mothers had a bachelor’s or master’s degree (N = 11) and were employed (N = 8). Persistent sleep and crying problems (N = 5), and persistent crying problems (N = 4) of the infant were the most common reasons for admission to the IMH day clinic. Four mothers were admitted to a crisis psychiatric ward during admission to the IMH day clinic. The infant’s mean age (SD) at the time of admission was 10.9 (5.2) months and the mean duration of admission was almost 6 (5.9, SD 2.0) months.

Analyses identified three major themes: (1) ‘experience of pregnancy, birth, and parenthood’; (2) ‘difficult care paths’; and (3) ‘fulfillment of needs’; and three subthemes: (1) ‘reality does not meet expectations’, (2) ’resilience under pressure’, and (3) ‘despair’. The interplay between the major themes and subthemes is shown in Figure 1 and is described in more detail below.

### 3.1. Experiences of Pregnancy, Birth and Parenthood, and Difficult Care Paths

All mothers had their expectations about pregnancy, childbirth, and the postpartum period influencing their experiences. Mothers pronounced the idea that motherhood comes from a natural-born instinct. Some mothers expected parenthood with the associated care responsibilities to come naturally (for example, breastfeeding). In addition, a few mothers felt that motherhood implies a certain level of (self)sacrifice. 


*“I have always thought that I’m going to have a child and nature will prepare you for it’.”*

*—M10*


Women described how pregnancy and childbirth were portrayed as moments of true happiness with no space for negative feelings. Maternity leave was expected to be a period of rest in which a mother can fully enjoy the time with her newborn. The perception of this ‘pink cloud’ was enhanced by the perception of the mother about existing opinions in society, shared through various channels like social media and television. 


*“If you read in magazines: people give birth and it’s a ‘pink cloud’ and everything goes well and ‘we are so happy with our baby’.”*

*—M10*


Expectations were also influenced by their own parenting experiences as a child. When mothers experienced a problematic parenting situation as a child, they were determined to be better parents for their children. 


*“I was convinced that someone who has had a difficult time as a child is well suited to do better for another child because you have more insight into the damage that this causes for a child. And I wish, yes, I wanted to show that I might be able to do it better.”*

*—M7*


Unfortunately, many women experienced the contrary. They often encountered a lack of understanding from their family and friends for not being on the ‘pink cloud’. Therefore, some mothers held up a facade and presented themselves in a way they thought others were expecting them to behave. 

“*And yes, then it just went downhill but I didn’t show it.”*
*—M6*


They acknowledged that this facade might be made it difficult for healthcare providers to note that their resilience was under great pressure and to detect mental health problems.


*“But I always pretended to be fine when they (cfr. midwife and maternity nurse) came to visit, so maybe this might be my own mistake because: you have your pride, it is your first child. You don’t want to admit that it’s actually not fine.”*

*—M9*


A few mothers, however, chose not to hold up this facade.


*“…and I never lied about my story. Because there are a lot of people who pretend to the world that it’s all a bed of roses, but I never did that.”*

*—M12*


The difficult care path, characterized by a lack of communication and referral also influenced mothers’ experiences during the perinatal period. These factors hindered their search for help. Often mothers only found out about the IMH facility by coincidence, for example, via an advertisement on (social) media or colleague. 


*“I saw on TV a mother being interviewed in the news, and she was talking about [infant mental health facility] […] and then I looked it up on the internet, I contacted them and that’s how I got an intake”*

*—M2*


Moreover, admission to the IMH facility came with a preexisting stigma, both in mothers as well as in healthcare providers. 


*“It was very threatening in the beginning. It’s baby psychiatry, that’s a big word. In our environment, everyone said: ‘a baby to a psychiatrist? What will they do? Chat with it?’.”*

*—M3*


Mothers also mentioned several pitfalls regarding the accessibility of mental healthcare services. A lack of connection between them and the healthcare provider, long waiting lists, and high costs associated with mental healthcare were discouraging to seek help.


*“And then I said, with tears in my eyes: ‘here it ends, this doesn’t work. I cannot afford that (cfr. psychotherapy)’.”*

*—M1*


### 3.2. Experiences of Pregnancy, Birth and Parenthood; and Fulfillment of Needs

Several factors challenged the mothers’ resilience. 


*“The lack of sleep, that was the biggest trigger for me. That completely undermined my resilience.”*

*—M5*


These factors, related to the experience of pregnancy, childbirth, and the postnatal period, led to negative emotions, contradicting the expected positive emotions associated with pregnancy and childbirth.

Almost all mothers felt the need to be heard, without judgment. 


*“I screamed for help but no one, no one heard me, no one really listened.”*

*—M12*


A supportive social network was important for the fulfillment of this need. The partner was the most important support figure. The absence of a partner, an unstable relationship, a lack of communication, or a lack of support regarding existing problems with the mother or the child, challenged the mothers’ resilience. 


*“I felt completely, well, very little supported by my partner. Uhm, we’ve almost stopped talking to each other.”*

*—M13*


Family members, especially their mother and friends, gave emotional and practical support. However, the well-intended but contradicting advice given by this network could also be perceived as disturbing and frustrating. Another support source was the work environment of the mother. A supportive working environment was characterized by flexibility and acceptance, for example, for taking parental leave. The experience of the contrary, receiving little or no support, led to feelings of loneliness and isolation.


*“I really felt completely isolated from the world, I felt completely misunderstood.”*

*—M6*


Another high valued form of support was peer support. The possibility to discuss their experiences with peers had a positive impact. Contrary to their network, peers were more understanding and acknowledged their feelings. Some of the participants are still in contact with the parents they met through the IMH facility.


*“Yes, I have noticed that I had a lot of support just by chatting with other moms, who also experienced similar situations (…)”*

*—M11*


In addition to the need for support, mothers indicated that they had to learn to trust themselves and their competence again. Mothers described feeling disconnected. They no longer relied on themselves nor their gut feeling. As a result, they were unable to connect with their child, both physically and emotionally, causing a disturbed mother-child relationship. This disturbance in the relationship was also reflected in the child’s behavior, for instance, by looking away from the mother.


*“When I was admitted (cfr. to a crisis psychiatric ward), I really had the feeling ‘I don’t want that child, I really don’t. I, I would prefer to give him up for adoption (…)’”*

*—M10*


Getting confirmation from their environment and being encouraged by healthcare providers, was very important in this context. A prescribed sick leave by the IMH facility gave mothers the feeling of being allowed to rest. The prescribed rest and the mindfulness exercises helped them to reconnect with their intuition. Interventions to reconnect and rediscover the bond with their infant were also experienced as helpful.


*“That [crf. baby dancing] was one of the few moments that we really enjoyed each other. (…) Because it was so intense to feel that connection with him. I really needed that to feel like ‘oh, it’s there. It is just very difficult between us for a while, but it is there’.”*

*—M6*


The individual psychological sessions with a counselor at the IMH facility were also perceived as supportive. 


*“We just got a lot of information at the infant mental health facility, that was something to hold on to and an explanation for certain situations”*

*—M1*


During their admission to the IMH facility, in a few cases, the resilience of the mother was under pressure to such an extent that inpatient admission of the mother to a psychiatric hospital was necessary, to guarantee the safety of both the mother and her child. Although the mothers initially expressed resistance, later they felt relieved. By stepping out of the family environment and allowing themselves to rest, mothers were given the chance to get out of the negative spiral.


*“Yes, the negative spiral of ‘I’m a bad mother … and there is no reason for me to carry on like this’. That really was my attitude.”*

*—M10*


After discharge, mothers felt the need for maintaining professional support in case of questions or concerns.


*“I have always had the feeling that the infant mental health facility provided a safety net after our discharge.”*

*—M6*


### 3.3. Difficult Care Paths and Fulfillment of Needs

As mentioned above, mothers felt the need to be heard. However, mothers felt that some healthcare providers took a paternalistic approach, making them not feel taken seriously. Mothers felt frustrated and, in some cases, they stopped the care process. 


*“But [sighs] most doctors always stay above you (…).”*

*—M5*


The absence of professional support caused despair in mothers. A sense of failure as a mother, as a partner, and as an employee, emphasized this feeling of despair affecting the mother’s self-image negatively. Mothers felt guilty and blamed themselves for the situation they were in.


*“I felt like the worst mother ever because what kind of mother can’t comfort her own child?”*

* —M9*


Some mothers lost their selves in their work and some cases, in narcotics, such as sleep medication.


*“Uhm, but I lost myself in my job…”*

* —M6*


Nonetheless, mothers reported a determination to find a cause and a solution for their infant’s excessive crying, sleeping, and/or feeding problems. Therefore, mothers visited many different healthcare providers during the postpartum period. The general practitioner often played a passive role, despite the fact that he or she sometimes acted as a confidential counselor for the mother. The postpartum care by the midwife was rather short-term and the problems often arose after midwifery care was over (approximately 6 to 12 weeks postpartum). All mothers consulted a pediatrician who typically tried to exclude physical problems in the child (e.g., prescription of drugs like omeprazole, tests to exclude cow’s milk protein intolerance). If this did not improve the situation, no alternatives were left. The lack of referral to specialized mental healthcare after all other medical approaches failed left mothers with a feeling of not being heard nor understood.

“*I only encountered a lack of understanding in the medical world and that frustrates me the most …”*
*—M11 *


## 4. Discussion

This study aimed to retrospectively explore the experiences and needs of mothers regarding professional support prior to their admission to an infant mental health day clinic because of persistent severe infant regulatory problems. Three major themes were identified: ‘experience of pregnancy, birth, and parenthood’, ‘difficult care paths’, and ‘needs and their fulfillment’.

The transition into parenthood is a complex interaction of stressors and resources, in which finding a balance between these two is crucial. The extent to which parents are capable of finding this balance is captured in the literature as resilience. Resilience in pregnancy and after birth is multi-factorial and influenced by individual, socio-cultural, and environmental factors [28]. Van Haeken et al. (2020) conducted a concept analysis and Delphi survey of perinatal resilience which found it can be understood as a circular process towards greater wellbeing in the form of personal growth, family balance, adaptation, or acceptance when faced with stressors, challenges, or adversity during the perinatal period. It consists of five main attributes: social support, sense of mastery, self-efficacy, self-esteem, and personality, which enhance the capacity of women to be resilient [29]. The mothers’ narratives indicate that the combination of different stressors put the main perinatal resilience attributes under pressure. Women’s sense of mastery was reduced because they no longer felt they controlled the situation and lost connection with their feelings and with those of their children. Mothers no longer believed in their abilities to cope with motherhood and did not perceive themselves as “good” mothers, leading to a negative self-image and poor self-esteem. Another important pitfall was the lack of understanding and adequate social support from informal and formal support networks. Mothers consulted various healthcare providers because of persistent severe infant regulatory problems, but mothers felt not being taken seriously, which undermined their sense of mastery and self-efficacy. Mothers found the recognition and confirmation of their feelings and experiences mainly from their partners and peers at the infant mental health facility. Peer support can be very helpful, but this only applies to peer support in which the experiences are similar. Otherwise, peer support might not be helpful [30]. In building resilience, social support is an important factor. Lack of social support has a significant impact on the well-being of women during and after pregnancy. Social support can be provided by different partners, each partner and its potential impact requires additional research.

Due to challenging factors related to pregnancy, childbirth, or the postpartum period, women experienced negative feelings contradicting the anticipated feelings of happiness. The majority of the mothers had expectations influenced by their social environment and the society in general, that were not met. This can be the result of a knowledge deficit as was shown in a Canadian study with first-time parents, in which only 44% reported the feeling of being adequately prepared for parenthood [31]. Similarly, the systematic review of Scope et al. (2017) [20] showed that women experienced a lack of knowledge about coping and caring for a newborn, but that they only became aware of this lack after following a postnatal intervention to prevent postpartum depression. Realistic information about common experiences can promote more informed expectations about the changes that come with the transition to parenthood, and thus potentially decrease distress [32].

In Belgium, the perinatal healthcare system is characterized by a variety of professions (obstetricians, midwives, general practitioners, and pediatricians) organized in hospitals as well as in private care which provides a multidisciplinary healthcare setting. However, the fragmented care provision makes access to services complex for future parents and complicates communication and referral between healthcare providers [6,33]. In our study, the lack of communication among healthcare providers and a lack of continuity of care potentially complicated appropriate referral. As a result, mothers may drop out of the care pathway before receiving appropriate care [34]. In this study, the midwife, the general practitioner, and the pediatrician were most often involved in postnatal care, but existing problems were not adequately recognized. Postnatal midwifery care often was finished when the problems became apparent. The general practitioner adopted a ‘watch and wait’ approach in the screening, detection, or referral to specialized mental healthcare, comparable to findings of Ford et al. (2017) [34]. Almost all participants consulted a pediatrician more than once. The pediatrician interacted frequently with the mother after midwifery care was ended [35]. Therefore, the pediatrician can play an important role in detecting infant mental health problems [19,36]. While research indicates that pediatricians feel responsible for detecting postpartum disorders, such as maternal postpartum depression, they are among the physicians least likely to screen for postpartum depression, hindering adequate referral [35]. In line with the study of Fowles et al. (2012) [37], in our study women reported that professionals mainly focused on the medical conditions of the infant, preventing a broader approach towards the infant’s regulatory problem. Less attention was paid to the mother-child dyad and no attention to the family triad. Recognition of perinatal mental health problems seems to be hindered by the existing stigma regarding mental health in mothers and healthcare providers [6].

However, the perinatal period provides a window of opportunity in which healthcare providers may screen for resilience-related problems and detect a decline in resilience attributes, ideally starting from prenatal consultations [33,38]. In line with previous research, we find that a decrease in perinatal resilience attributes such as sense of mastery [39], social support [40], self-efficacy [41], and self-esteem [42] can be a precursor of perinatal mental health problems. Thus, adapting a resilience-oriented approach into existing prevention strategies focusing on enhancing perinatal resilience potentially stimulates addressing, detecting, and treating mental health problems.

### Strengths and Limitations

This study provides an in-depth understanding from the perspective of mothers on the factors that challenge the mother-child relationship and perinatal resilience. The approach is bottom-up, providing direct insights from the mothers, and provides greater insight into the lived experiences of women compared to quantitative designs. Despite these strengths, some limitations need to be considered. First, recruitment of participants took place in only one infant mental health facility in Belgium, generalization of the results to other settings, therefore, warrants caution. Second, we only included mothers. Therefore, the perspective of partners is missing which is relevant for a broader understanding of the supportive environment of women. Third, the sample consists mainly of Caucasian, highly educated, employed women who were living with their partners. It is therefore important that further research is initiated to examine the experiences and needs of mothers and their partners from more diverse backgrounds. Fourth, the main focus of the interviews was on the experiences and needs regarding professional support prior to admission to an IMH facility, with the consequence that healthcare during pregnancy and the experience of childbirth was not systematically questioned. Future research can include the family triad as well as the experiences during pregnancy and birth to obtain a more holistic perspective.

## 5. Conclusions

The mothers in our sample struggled to find a balance between the stressors they experienced during the perinatal period and their resources, putting their perinatal resilience under pressure. Mothers experienced a sense of failure, a disconnection with themselves and their infant, and did not believe in their abilities to cope with motherhood. A lack of social support from their (in)formal network made women feel isolated. The negative feelings women experienced were in contradiction to the expected positive emotions associated with childbirth and motherhood. This negatively affected the mother-child relationship, and infants’ regulatory capacities. Determined to find solutions for the latter, different healthcare providers were consulted. Mothers’ search for help, however, was complex and communication between healthcare providers was limited because of a fragmented care provision. This hindered the continuity of care and appropriate referrals. Another pitfall was the lack of a holistic approach, with the emphasis on the medical aspects without attention to the mother-child dyad. From this study, the following can be learned: (1) mental health needs to be a component of an integrated care pathway focusing on early detection of resilience-related problems; (2) social support should be addressed in perinatal care as a key attribute to support maternal resilience; and (3) integrated care should be provided with the family triad as a focus, up to the child’s age of two. Last, societal sensibilization regarding the importance of perinatal mental health and how to support new parents should be a part of public health campaigns.

## Figures and Tables

**Figure 1 ijerph-18-10917-f001:**
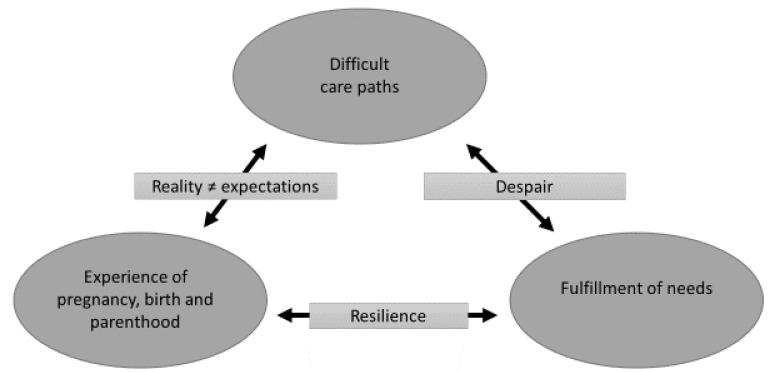
The interplay between the major themes and subthemes.

**Table 1 ijerph-18-10917-t001:** A semi-structured interview guide, including icebreaker questions.

Questions About Experiences and Needs Regarding Professional Support
Can you tell me something about yourself and your family?
Can you tell me something about the time before you were admitted to the IMH day clinic? How did you experience this period? How did you experience admission at the IMH day clinic? What did the admission do to you? After you and your child were discharged from the IMH day clinic, how did you experience this period (focus is up to 2 years after pregnancy)? What does perinatal resilience mean to you?

**Table 2 ijerph-18-10917-t002:** Socio-demographic and IMH day clinic characteristics of the mothers and their families.

Characteristics	Mothers (N = 13)	Fathers ^1^ (N = 11)
Ethnicity (n)		
Caucasian	13	11
Mean age, in years (SD) *	33.6 (4.3)	34.3 (2.8)
Education level (n)		
Until 18 years	2	6
Bachelor/Master degree	11	5
Employment * (n)		
Employed	8	11
On sick/maternity leave	5	0
Student	1	0
Parity at admission (n)		
Primiparous	8	
Multiparous	5	
Number of children (median, range)	2 (1–3)	
Type of regulatory problem of infant (n)		
Persistent sleep and crying problem	5	
Persistent crying problem	4	
Persistent sleep problem	1	
Persistent eating problem	1	
Persistent crying and eating problem	1	
**Persistent sleep and crying and eating problem**	1	
Admission of the infant to IMH day clinic		
**Mean age in months (SD)**	10.9 (5.2)	
Readmission (n)	2	
**Mean total duration of admission, in months (SD)**	5.9 (2.0)	
Admission to crisis psychiatric ward during infant’s admission to IMH facility (n)	4	NA
Medication use * (n)		
Antidepressants	4	NA
Other **	3	NA

^1^ Socio-demographic characteristic reported by the mothers as partners were not interviewed for this study, * at the moment of the interview, ** non-psychotropic medication.

## Data Availability

Coded transcribed interviews of the study can be retrieved through the corresponding author on reasonable request.

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
