# Peer review of "“Nobody Listened”. Mothers’ Experiences and Needs Regarding Professional Support Prior to Their Admission to an Infant Mental Health Day Clinic"

_ijerph, 2021, doi:10.3390/ijerph182010917_

Round 1

Reviewer 1 Report

The authors study the experiences and needs regarding professional support of mothers with difficulties in mother-infant interaction prior to their admission to an infant mental health day-clinic. This is a relevant and interesting topic. I have some minor suggestions for your manuscript.

Keywords could add and complement the title. Consider using MeSH terms: "Delivery of Health Care, Integrated"; "Holistic Health"

Introduction:

The rationale for the study was well explained. However, a more comprehensive and more critical overview of public health approaches to perinatal mental health would be expected in the introduction. Your introduction could better link perinatal mental health and the development of supportive environments in building resilience. Social determinants are also important causes of mental health problems in pregnant women and mothers which are often overlooked.

I would recommend the following articles:

Blount, A. J., Adams, C. R., Anderson-Berry, A. L., Hanson, C., Schneider, K., & Pendyala, G. (2021). Biopsychosocial Factors during the Perinatal Period: Risks, Preventative Factors, and Implications for Healthcare Professionals. International journal of environmental research and public health, 18(15), 8206. https://doi.org/10.3390/ijerph18158206

Tripathy P. A public health approach to perinatal mental health: Improving health and wellbeing of mothers and babies. J Gynecol Obstet Hum Reprod. 2020 Jun;49(6):101747. doi: 10.1016/j.jogoh.2020.101747.

Viveiros CJ, Darling EK. Perceptions of barriers to accessing perinatal mental health care in midwifery: A scoping review. Midwifery. 2019 Mar;70:106-118. doi: 10.1016/j.midw.2018.11.011.

Methods: Methods are very clear and well described. Consider given more details concerning the data collection process.

How was the study presented to the participants? Also clarify the rationale for the number of interviews performed (all of the eligible women who accepted to participate n=13: 9 + 4 for saturation?).

Please discuss ethical considerations beyond participants’ confidentiality based on existing guidelines, namely those related to reducing participant distress caused by the research.

Results:  

Were all mothers and fathers born in Belgium? Was other information on socioeconomic status besides education collected? And also data about the type of birth and healthcare during pregnancy?  

Results could better reflect participants' own voices and experiences, namely if quotations presented some additional information about the social characterization of the participants and parity. Also, although the sample consists mainly of Caucasian and employed mothers living with their partners, some variation could be explored considering the perception of social support / enabling a better understanding of how to create supportive environments to promote perinatal mental health.

Discussion:

The authors could better detail the well-identified study limitations and discuss their implications for the results.

Conclusion:

Although results are exploratory, implications of the results for public policy should be enlightened with the identification of good practices.

Author Response

Point by point response to the comments and recommendations made by the reviewers:

We thank the reviewers for recognizing the added value and importance of the topic, as well as their suggestions and additional views on professional support of mothers with difficulties in mother-infant interaction and the relevance for the public health area. Below we address the minor suggestions to further improve the manuscript. 

Reviewer 1

The authors study the experiences and needs regarding professional support of mothers with difficulties in mother-infant interaction prior to their admission to an infant mental health day-clinic. This is a relevant and interesting topic. I have some minor suggestions for your manuscript.

  1. Keywords could add and complement the title. Consider using MeSH terms: "Delivery of Health Care, Integrated"; "Holistic Health"

We would like to thank the reviewer for this suggestion. We included the keywords and adjustments can be found in the manuscript on line 45: Keywords: infant mental health; perinatal mental health; resilience; infant-parent interaction; healthcare organization; regulatory disfunction, “holistic health, delivery of health care integrated”

  1. Introduction:

The rationale for the study was well explained. However, a more comprehensive and more critical overview of public health approaches to perinatal mental health would be expected in the introduction. Your introduction could better link perinatal mental health and the development of supportive environments in building resilience. Social determinants are also important causes of mental health problems in pregnant women and mothers which are often overlooked.

I would recommend the following articles:

Blount, A. J., Adams, C. R., Anderson-Berry, A. L., Hanson, C., Schneider, K., & Pendyala, G. (2021). Biopsychosocial Factors during the Perinatal Period: Risks, Preventative Factors, and Implications for Healthcare Professionals. International journal of environmental research and public health, 18(15), 8206. https://doi.org/10.3390/ijerph18158206

Tripathy P. A public health approach to perinatal mental health: Improving health and wellbeing of mothers and babies. J Gynecol Obstet Hum Reprod. 2020 Jun;49(6):101747. doi: 10.1016/j.jogoh.2020.101747.

Viveiros CJ, Darling EK. Perceptions of barriers to accessing perinatal mental health care in midwifery: A scoping review. Midwifery. 2019 Mar;70:106-118. doi: 10.1016/j.midw.2018.11.011.

We thank the reviewer for this relevant suggestion. In line with the reviewers suggestion, we have added the importance of a supportive environment in the prevention of perinatal mental health problems to the introduction as well as the suggested references. Changes can be found in line 79-82: “Mothers have stressed the importance of a supportive environment. And a trusting therapeutic relationship as a key factor in creating a safe space where their needs can be met.”
The importance of a supportive environment in building resilience is discussed in detail within the discussion of the manuscript.

  1. Methods: Methods are very clear and well described. Consider given more details concerning the data collection process.

How was the study presented to the participants? Also clarify the rationale for the number of interviews performed (all of the eligible women who accepted to participate n=13: 9 + 4 for saturation?).

As mentioned in line 124, eligible women received study information by an IMH facility staff-member. When women were interested to participate to the study, they were contacted by one of the researchers for further details about the study protocol. The research team that conducted data collection and analysis was independent of the IMH facility members. In total 13 women were interviewed. Data saturation, this means the moment where no new information is obtained, was reached after 9 interviews. In order to confirm saturation, four more interviews were conducted after which no new themes emerged. Additional clarification was added in line with the reviewer's comment (line 166-168); “The process was continued until data saturation was reached; no new information was obtained from subsequent interviews and thus recruitment was stopped.”

Please discuss ethical considerations beyond participants’ confidentiality based on existing guidelines, namely those related to reducing participant distress caused by the research.

To address the relevant comment of the reviewer, we have clarified this under the heading of the ethical considerations as follows; “Participants took part voluntarily and had the opportunity to discontinue their participation at any time during the interview and study process. In case of any distress the IMH service provided back up.” (line 116-118)

  1. Results:  

Were all mothers and fathers born in Belgium? Was other information on socioeconomic status besides education collected? And also data about the type of birth and healthcare during pregnancy?  

We understand the comment of the reviewer. We have therefore added the birthplace, Belgium, to the results (line 173). Besides, education, employment, parity, number of children, types of regulatory problem of the infant, medication use and information about admission to the IMH day-clinic were assessed as shown in table 2. Type of birth was not assessed. The main focus of the interview guide was on the experiences and needs regarding professional support prior to admission to an infant mental health day-clinic starting after childbirth. For this reason the manuscript does not contain information about the (experiences of) healthcare during pregnancy and birth. We agree with the reviewer this may have been of interest, we have included this in the limitations of the study as follows: “Fourth, the main focus of the interviews was on the experiences and needs regarding professional support prior to admission to an IMH facility, with the consequence that healthcare during pregnancy and the experience of childbirth were not systematically questioned. Future research can include the family triad as well as the experiences during pregnancy and birth to obtain an even more holistic perspective.” (line 422-427).

Results could better reflect participants' own voices and experiences, namely if quotations presented some additional information about the social characterization of the participants and parity. Also, although the sample consists mainly of Caucasian and employed mothers living with their partners, some variation could be explored considering the perception of social support / enabling a better understanding of how to create supportive environments to promote perinatal mental health.

The reviewer's comment is certainly relevant regarding how to create supportive environments to promote perinatal mental health. However, after discussion within the research team, we decided not to add additional information about the participants other than mentioned in table 2 given the sensitivity of the topic and the recruitment from one IMH facility in Belgium, we want to guarantee as much anonymity as possible by not adding personal details to the quotes. We agree that social support is an important factor in building resilience and therefore added clarification in the discussion (line 362-366); In building resilience, social support is an important factor. Lack of social support has a significant impact on the wellbeing of women during and after pregnancy. Social support can be provided by different partners, each partner and its potential impact requires additional research[7].

  1. Discussion:

The authors could better detail the well-identified study limitations and discuss their implications for the results.

In line with the recommendation of the reviewer, we added details to the study limitations as follows: (1) Therefore, the perspective of partners is missing which is relevant for a broader understanding of the supportive environment of women. (line 418-419)
(2) Fourth, the main focus of the interviews was on the experiences and needs regarding professional support prior to admission to an IMH facility, with the consequence that healthcare during pregnancy and the experience of childbirth were not systematically questioned. Future research can include the family triad as well as the experiences during pregnancy and birth to obtain an even more holistic perspective. (line 422-427)

  1. Conclusion:

Although results are exploratory, implications of the results for public policy should be enlightened with the identification of good practices.

We thank the reviewer for the relevant remarks. Due to the adjustments, we have been able to optimise the manuscript. In response to this last comment, minor adjustments were made to the conclusion as follows: From this study the following can be learned: 1) mental health needs to be a component of an integrated care pathway focusing on early detection of resilience-related problems; 2) social support should be addressed in perinatal care as a key attribute to support maternal resilience; and 3) integrated care should be provided with the family triad as a focus, up to the child’s age of two. Last, societal sensibilization regarding the importance of perinatal mental health and how to support new parents should be a part of public health campaigns. (line 442-450).

Reviewer 2 Report

The paper presents an important topic to the public health area, whose relevance is clearly sustained across the introduction. Nevertheless, for the reasons detailed below is in need some minor work before its publication.

1) In the line 76, the study’s pertinence was justified by the fact that previous research was mainly focused on the mothers’ own mental health problems and less on the infant’s regulatory problems. Could you briefly detailed the main conclusions from these previous studies and how your results are expected to be different/similar from that?

2) In the line 89, is written that the interviews explored the experiences and needs of the participants, which is a little vague. Could you please be more specific about what experiences and needs are you writing about?

3) In the data analysis section is not clear what procedures were used regarding the grounded theory approach. Including, in line 132 is reported another data analysis' method (QUAGOL). How both were combined?

4) Could you find differences in your results according to the participants' socio-demographic characteristics? For example, when is writing "a few mothers" or "from some mothers", what distinct them from the others? Thinking in a more personalizaed health care intervention, it could be important to highligth what themes were reported by all/the majority of mothers and what are more specific from a particular group of them. 

Hoping that the presented suggestions help to improve the paper and congratulations to the authors to raise an important topic, with medium/long term implications, but that sometimes continues to be a public taboo (as some of "your" mothers ended up to assume). 

Author Response

Point by point response to the comments and recommendations made by the reviewers:

We thank the reviewers for recognizing the added value and importance of the topic, as well as their suggestions and additional views on professional support of mothers with difficulties in mother-infant interaction and the relevance for the public health area. Below we address the minor suggestions to further improve the manuscript. 

Reviewer 2

The paper presents an important topic to the public health area, whose relevance is clearly sustained across the introduction. Nevertheless, for the reasons detailed below is in need some minor work before its publication.

  1. In the line 76, the study’s pertinence was justified by the fact that previous research was mainly focused on the mothers’ own mental health problems and less on the infant’s regulatory problems. Could you briefly detailed the main conclusions from these previous studies and how your results are expected to be different/similar from that?

We thank the reviewer for this comment. We added the main conclusions from literature on line 79-82 as follows: “Mothers have stressed the importance of a supportive environment. And a trusting therapeutic relationship as a key factor in creating a safe space where their needs can be met. Perinatal mental health services were considered critical to address the relationship with their infant [19].

  1. In the line 89, is written that the interviews explored the experiences and needs of the participants, which is a little vague. Could you please be more specific about what experiences and needs are you writing about?

We agree to the reviewer's comments. Therefore, we have modified line 97 by adding the sentence: 'regarding professional support prior to their admission to an infant mental health day-clinic'.

  1. In the data analysis section is not clear what procedures were used regarding the grounded theory approach. Including, in line 132 is reported another data analysis' method (QUAGOL). How both were combined?

We understand the reviewer's confusion and have therefore clarified the data analysis process in the manuscript. QUAGOL is a method developed by researchers at KU Leuven and is a guide to facilitate the process of qualitative data analysis. This method is inspired by the constant comparative method of the grounded theory approach. We therefore added to the manuscript as follows: An exploratory qualitative research design was adopted, using the Qualitative Analysis Guide of Leuven (QUAGOL). This method is inspired by the grounded theory approach. This method was chosen to reconstruct the story of the participants on a theoretical level and to analyze the concepts found” (line 92-95)

  1. Could you find differences in your results according to the participants' socio-demographic characteristics? For example, when is writing "a few mothers" or "from some mothers", what distinct them from the others? Thinking in a more personalized health care intervention, it could be important to highlight what themes were reported by all/the majority of mothers and what are more specific from a particular group of them. 

The reviewer's comment is certainly relevant regarding specifying particular groups. However, after discussion within the research team, we decided not to add additional information about the participants other than mentioned in table 2 given the sensitivity of the topic and the recruitment from one IMH facility in Belgium, we want to guarantee as much anonymity as possible by not adding personal details to the quotes.

Hoping that the presented suggestions help to improve the paper and congratulations to the authors to raise an important topic, with medium/long term implications, but that sometimes continues to be a public taboo (as some of "your" mothers ended up to assume).